# Predictive Factors of Cytomegalovirus Viremia during the Clinical Course of Anti-Neutrophil Cytoplasmic Antibody (ANCA)-Associated Vasculitis: A Single Center Observational Study

**DOI:** 10.3390/jcm12010351

**Published:** 2023-01-02

**Authors:** Makoto Harada, Ryohei Iwabuchi, Akinori Yamaguchi, Daiki Aomura, Yosuke Yamada, Kosuke Sonoda, Yutaka Kamimura, Koji Hashimoto, Yuji Kamijo

**Affiliations:** Department of Nephrology, Shinshu University School of Medicine, 3-1-1, Asahi, Matsumoto 390-8621, Japan

**Keywords:** anti-neutrophil cytoplasmic antibody (ANCA), anti-neutrophil cytoplasmic antibody (ANCA)-associated vasculitis, cytomegalovirus, cytomegalovirus infection, cytomegalovirus viremia

## Abstract

We aim to elucidate factors to aid in the prediction of cytomegalovirus viremia during the treatment of anti-neutrophil cytoplasmic antibody-associated vasculitis (AAV). We conducted a single-center, retrospective, observational study of 35 patients with newly diagnosed AAV. Factors associated with the development of CMV viremia were investigated via a logistic regression analysis. The CMV antigenemia test was performed in 25 patients (71%), of whom 15 (60%) were diagnosed with CMV viremia. Of these 15 patients, 5 developed a CMV infection. The total protein, hemoglobin, platelet count and lymphocyte counts at the time of the CMV antigenemia test were significantly lower in patients who developed CMV viremia. In addition, total protein, hemoglobin, platelet count and lymphocyte count also presented significantly decreasing trends in the following order: patients who did not develop CMV viremia, patients who developed CMV viremia without any symptoms, and patients who developed CMV infection. All patients with CMV recovered. In conclusion, the total protein, hemoglobin, platelet count and lymphocyte count may be useful markers for the prediction of CMV viremia and infection after the start of induction of immunosuppressive therapy for patients with AAV.

## 1. Introduction

Anti-neutrophil cytoplasmic antibody (ANCA)-associated vasculitis (AAV) is a systemic small vessel vasculitis that leads to severe kidney dysfunction and/or lung injuries such as alveolar hemorrhage and interstitial pneumonitis [1,2]. As the pathogenesis of AAV is associated with autoimmune disorders, immune suppression is often used to treat AAV [1,2]. Rituximab and/or cyclophosphamide and glucocorticoid therapy are generally administered for AAV induction therapy, and azathioprine and/or glucocorticoids are administrated for maintenance therapy [1,2]. AAV often develops in elderly patients, and infectious complications are one of the most important problems in the therapeutic course of AAV [3,4] as the most common cause of death in patients with AAV [4,5]. In addition, opportunistic infections such as cytomegalovirus (CMV) infection, tuberculosis, and pneumocystis pneumonia have been reported in patients undergoing treatment for AAV [4,5,6]. CMV is a herpes virus with which most people are latently infected [6,7,8]. With the immune system weakened during AAV treatment, CMV reactivates and proliferates, resulting in organ impairments such as pneumonitis, hepatitis, enterocolitis, and retinitis as well as hematological disorders [7]. Ganciclovir and valganciclovir are the standard treatments for CMV infections [7,8]. The CMV antigenemia test evaluates the CMV-pp65-antigen-positive cell count and is useful for the early detection of CMV viremia and infection [8]. Routine CMV antigenemia tests are performed for patients who have undergone solid organ or hematopoietic cell transplantation [8].

Several case reports have demonstrated that severe infectious conditions due to CMV infections occur during the therapeutic course of AAV [9,10,11]. However, effective prediction and treatment methods for CMV viremia or infection during treatment for AAV have yet to be established. The European League Against Rheumatism, guidelines from the British Society of Rheumatology, and a nationwide survey in Japan do not address the issue of CMV viremia or infection in patients undergoing treatment for AAV [3,12,13]. One ongoing clinical study is investigating the efficacy of prophylactic treatment against CMV in patients with AAV; however, the results have yet to be reported [14]. Lim et al. have addressed the problem of the cost-effectiveness of prophylactic management against CMV infection in patients with AAV [15]. Overall, although CMV infections during treatment for AAV are a significant infectious complication, there is little information on how to predict and treat this condition. This study aimed to identify factors associated with CMV to aid with the prediction of CMV viremia, which is considered the early phase of CMV infectious complications. Determining the predictive factors of CMV viremia may lead to the early detection and intervention of CMV infections or an adjustment in the strength of immunosuppressive therapy.

## 2. Materials and Methods

This is a single-center, retrospective, observational study of patients who were newly diagnosed with AAV at the Department of Nephrology, Shinshu University Hospital, between January 2013 and December 2019. Patients younger than 18 years or those who had been treated with immunosuppressive therapy prior to AAV induction therapy were excluded. In addition, patients positive for glomerular basement membrane (GBM) antibody in addition to ANCA (myeloperoxidase (MPO) and/or proteinase 3 (PR3)) were excluded, as these patients may receive stronger immunosuppressive therapy. Of 39 patients eligible for this study, 2 were excluded due to anti-GBM positivity, and 2 were excluded due to treatment at an outside hospital (Figure 1). The final analysis included 35 patients. This study was approved by the institutional review board of the ethical committee at Shinshu University School of Medicine (approval number: 4809) and was conducted in accordance with the principles of the Declaration of Helsinki. The requirement of written informed consent was waived due to the retrospective nature of the study.

Thirty-nine patients were eligible for inclusion. Two patients were excluded due to positivity for GBM antibody, and two were excluded due to treatment at an outside hospital. The final analysis included 35 patients, of which 25 received the CMV antigenemia test, and 15 were diagnosed with CMV viremia.

AAV was defined according to the algorithm suggested by Watts et al. [16]. We categorized each patient as eosinophilic granulomatous with polyangiitis (EGPA), granulomatous polyangiitis (GPA), microscopic polyangiitis (MPA), or unclassifiable. 

Diabetes mellitus was defined as a high level of HbA1c (>6.5%), an insulin or hypoglycemic agent prescription, and/or a history of diabetes mellitus listed in the patient’s medical records. Hypertension was defined as antihypertensive drugs prescription and/or a history of hypertension described in the medical records. Interstitial pneumonitis was defined as bilateral interstitial lesions on computed tomographic images. Alveolar hemorrhage was defined as hemoptysis and lung abnormalities on computed tomography that corresponded to hemorrhage. Neurological symptoms were defined as numbness and muscle weakness.

The maximum dose of prednisolone (PSL) was adjusted according to ideal body weight. A high dose of PSL was defined as patients who received PSL more than the median dose. In a previous Japanese nationwide study, rapid PSL reduction was defined as when the necessary daily PSL dose decreased to <20 mg/day within 8 weeks [3]. Methylprednisolone pulse therapy consisted of 500–1000 mg/day of intravenous methylprednisolone administered for 3 consecutive days. Cyclophosphamide was administered intravenously. 

The severity of AAV was evaluated according to the Birmingham vasculitis score 3 (BVAS-3) at the time of hospital admission [17]. AAV treatment was performed according to the Japanese guideline of ANCA-positive rapid progressive glomerulonephritis [3]. Blood and urine samples obtained at the time of hospital admission and the CMV-pp65-antigen-positive cell count were evaluated. The CMV-pp65-antigen-positive cell count was evaluated after the start of induction of immunosuppressive therapy via the antigenemia method (LSI Medience, Corporation, Tokyo, Japan). CMV viremia was defined as the detection of at least one CMV-pp65-antigen-positive cell. The timing of the CMV antigenemia test was determined by the treating physicians and was typically performed when the patient was determined to be at high risk of CMV viremia or infection due to treatment with strong immunosuppressive therapy or when the patient was suspected to have developed a CMV infection. However, there were no strict guidelines regarding the timing of the CMV antigenemia test. The timing of anti-CMV treatment was also determined by the treating physicians, according to the CMV-pp65-antigen-positive cell count.

Clinical data were collected from patients’ medical records. CMV infection was defined as fever, lung lesion, liver dysfunction, abdominal symptoms, eye symptoms, and/or a hematological disorder due to CMV. Severe kidney dysfunction that required maintenance dialysis therapy was defined as end-stage renal disease at the time of hospital discharge. Proteinuria was defined as urinary protein > 0.15 g/gCr. Hematuria was defined as a red blood cell sediment count > 5/high power field. Anti-CMV-IgG was evaluated by the chemiluminescent enzyme immunoassay method (LSI Medience, Corporation, Tokyo). CMV seropositivity was defined as an anti-CMV IgG titer > 6.0 AU/mL, and a titer > 250 AU/mL was considered high.

Continuous variables are presented as the median and range and categorical variables as the number (n) and percentages (%). Continuous variables were compared via the Mann–Whitney U test, and categorical variables were compared via the Fisher’s exact test. We analyzed the association between age, male sex, body mass index, BVAS-3, serum IgG level, high titer of CMV-IgG level, and treatment pattern and development of CMV viremia using univariate logistic regression analysis. These factors were chosen based on previous reports, and the analyses revealed the relationship between patients’ baseline conditions or treatment pattern and the development of CMV viremia. The association between laboratory findings (total protein, serum albumin, blood urea nitrogen, serum creatinine level, aspartate aminotransferase, alanine aminotransferase, C-reactive protein, white blood cell count, hemoglobin level, and platelet count) and CMV viremia were also evaluated using logistic regression analysis. The factors that significantly associated with CMV viremia were evaluated using receiver operation curve analyses to detect the cutoff value associated with CMV viremia. The cutoff value was set at the maximum sum of sensitivity and specificity. The data of the patients who did not develop CMV viremia, who developed CMV viremia without any symptoms, and who developed CMV infection were evaluated using the Jonckheere–Terpstra trend test. A *p* value < 0.05 was considered statistically significant. Analyses were performed using EZR (Saitama Medical Center, Jichi Medical University, Saitama, Japan), which is a graphical user interface for R (The R Foundation for Statistical Computing, Vienna, Austria) [18], and IBM SPSS Statistics software package version 26 for Windows (IBM Co., Ltd., New York, NY, USA).

## 3. Results

### 3.1. Comparison of Background Data between Patients Who Received CMV Antigenemia Test or Not

The median patient age was 74 years, and the male-to-female ratio was 4:3. All patients were found to have MPO-ANCA, and 33 patients (94%) were categorized as MPA, 1 patient as EGPA, and 1 patient was unclassifiable. The median BVAS score was 18. As for the clinical symptoms, of 35 patients, no patients were detected to present obvious fever (>38 °C), and 8 patients had slight fever (38 °C>, ≥37 °C) at the time of hospital admission. Other symptoms such as purpura and neurological disorder are presented in the Table 1. The CMV antigenemia test was performed in 25 patients (71%) (Table 1). The maximum dose of PSL, blood urea nitrogen, and serum creatinine were significantly higher among patients who received the CMV antigenemia test compared to those who did not. Hemoglobin was significantly lower in patients who received the CMV antigenemia test (Table 1).

### 3.2. Comparison of the Background Data between Patients Who Developed CMV Viremia and Those Who Did Not

Of the 25 patients who underwent the CMV antigenemia test, 15 were positive (Table 2 and Figure 1). The patients’ clinical characteristics, laboratory data (at the time of hospital admission) treatment patterns, and clinical events are presented in Table 2. As for the clinical symptoms among patients who developed CMV viremia, they are asymptomatic except for patients who developed CMV infection. Patient age, history of hypertension and diabetes mellitus, disease severity (BVAS), treatment pattern, and incidence of ESRD or hospital death were not significantly different between patients with and without CMV viremia (Table 2). However, patients with CMV viremia tended to be treated by methylprednisolone pulse treatment or a higher maximum dose of prednisolone and had significantly higher levels of blood urea nitrogen, white blood cell counts and neutrophil counts (Table 2).

### 3.3. Association between Clinical Factors and Development of CMV Viremia

The univariate logistic regression analyses revealed that none of the analyzed factors were significantly associated with the development of CMV viremia (Table 3). The total protein, hemoglobin, platelet count and lymphocyte count at the time of CMV antigenemia test were significantly lower in patients who had CMV viremia than those who did not (Table 4). The duration between the start of immunosuppressive therapy and the CMV antigenemia test was not significantly different between the groups (Table 4). The total protein, hemoglobin, platelet count, and lymphocyte count were significantly associated with CMV viremia (Table 5).

### 3.4. Cutoff Values of the Prediction of CMV Viremia of Total Protein, Hemoglobin, Platelet Count and Lymphocyte Count

The cutoff values for the prediction of CMV viremia of total protein, hemoglobin, platelet count and lymphocyte count were 5.5 g/dL, 9.0 g/dL, 14.6 × 10^4^/μL and 750/μL, respectively (Figure 2). The area under the curves of combined total protein, hemoglobin, platelet count and lymphocyte count did not provide a more precise prediction method than any one variable alone (data not shown). The total protein, hemoglobin, platelet count and lymphocyte count at the time of the CMV antigenemia test presented significantly decreasing trends in the following order: patients who did not develop CMV viremia, patients who developed CMV viremia without any symptoms, and patients who developed CMV infection (Figure 3). 

The total protein, hemoglobin, platelet count, and lymphocyte count at the time of the CMV antigenemia test presented significantly decreasing trends in the following order: patients who did not develop CMV viremia (n = 10), patients who developed CMV viremia without any symptoms (n = 10), and patients who developed CMV infection (n = 5) (*p* values for the trends are as follows: serum total protein (*p* = 0.004), hemoglobin (*p* = 0.003), platelet count (*p* = 0.002) and lymphocyte count (*p* = 0.005)). 

### 3.5. Details of the Clinical Course of AAV Patients Who Developed CMV Viremia

Five patients who were diagnosed with CMV viremia did not require treatment, while the PSL dose was reduced in five patients, and ganciclovir or valganciclovir was administered to seven patients (Table 6). CMV pneumonitis, CMV colitis, and fever and anemia due to CMV infection were diagnosed in one patient each. Two patients presented with thrombocytopenia due to CMV infection. All patients improved throughout the study period.

## 4. Discussion

The total protein, hemoglobin, platelet count, and lymphocyte count obtained after the induction of immunosuppressive therapy for AAV are useful for predicting CMV viremia and/or infection. 

The prevalence of CMV viremia and/or infection, accurate prediction methods, and effective treatments for CMV-related infectious complications in patients undergoing treatment for AAV are not reported in major AAV guidelines [3,12,13]. A previous study suggested that almost all of the elderly population has anti-CMV-IgG [19], which is true of all patients in this study. Therefore, CMV reactivation and CMV viremia or infection may develop in nearly all patients with AAV. 

In the current study, CMV viremia was found in 15/25 patients who underwent the CMV antigenemia test, and the CMV-pp65-antigen-positive cell count was >5 in 7/25 (28%) patients. Morishita et al. detected a CMV-pp65-antigen-positive cell count > 5 in 13/111 (11.7%) patients with AAV [6]. The increased rate of high CMV-pp65-antigen-positive cell counts in this study may be due to the fact that the CMV antigenemia test was performed on patients with a high risk of CMV viremia and those who were suspected of having a CMV infection, resulting in a high pretest probability. Furthermore, patients with asymptomatic CMV viremia may not have undergone CMV antigenemia testing. Therefore, the actual prevalence of CMV viremia among patients with AAV is unclear, and further research is necessary.

More research is also necessary to identify risk factors of CMV viremia or infection so that prediction of CMV reactivation can occur. In addition, the most effective timing of the CMV antigenemia test also requires more research. Several studies have investigated CMV viremia in patients with autoimmune diseases [20,21]. Cui et al. reported that the prevalence of CMV antigenemia was higher in patients with systemic lupus erythematosus (SLE) than in those with other autoimmune diseases such as vasculitis and rheumatoid arthritis [20]. Kaneshita et al. reported that CMV was reactivated in 80/382 patients with autoimmune diseases [21]. It has been reported that AAV is the third most common background disease for CMV infection [21]. Oral candida infection, high CMV-pp65-antigen-positive cell counts, and hypoalbuminemia are possible risk factors for the progression of CMV viremia to CMV infection [21]. Shimada et al. reported that elderly, low level of serum albumin, higher creatinine level, cyclosporine use, and higher maximum and cumulative doses of PSL were the risk factors of CMV re-activation in rheumatic diseases [22]. Suga et al. reported that age > 60 years, lymphocytopenia (<1000/μL) and steroid pulse therapy were the risk factors of CMV infection in rhematic disease [23]. As for lymphocytopenia, we also detected as a predictive factor of CMV viremia, and its cutoff value was <750/μL in the current study. The use of rituximab therapy for AAV is becoming more frequent, however, it may lead to higher rates of CMV reactivation in patients with lymphoma or other hematological malignancies [24]. Lee et al. reported that rituximab therapy significantly increased CMV-related complications in patients diagnosed with non-Hodgkin lymphoma following autologous hematopoietic stem cell transplantation [25]. Although rituximab use was not significantly associated with the development of CMV viremia in this study, a future study with a large sample size is needed to investigate the association of rituximab and CMV reactivation or viremia in patients with AAV. No immunosuppressants used to treat patients in this study (maximum dose of prednisolone, methylprednisolone pulse, or cyclophosphamide) were significantly associated with the development of CMV viremia. A previous study suggested that GPA and disease severity may be associated with the development of CMV infection [6]. However, no patients were categorized as GPA in this study. Disease severity (BVAS-3) and treatment pattern were not significantly associated with the development of CMV viremia, while the maximum dose of prednisolone and the incidence of methylprednisolone use tended to be high in patients with CMV viremia. More research with a larger patient population is required to determine the effects of disease severity and treatment patterns on CMV reactivation and viremia. It is thought that a high anti-CMV-IgG titer is associated with the suppression of the development of CMV viremia; however, no significant association was detected in this study. More research with a larger patient population is required to determine the effects of a high anti-CMV-IgG titer. 

The results of this study are useful when determining the timing of detecting CMV viremia. On average, the CMV antigenemia test was performed 23.5 days (range: 9–36 days) after the start of induction of immunosuppressive therapy in patients who were detected to have CMV viremia and/or infection. In addition, the total protein, hemoglobin, and platelet count at the time of the CMV antigenemia test were significantly lower and found to be associated with CMV viremia. Immunosuppressive therapy decreases the serum immunoglobulin level, which may lead to a decrease in the total protein level, despite that serum albumin levels are unchanged. Unfortunately, serum immunoglobulin data at the time of the CMV antigenemia test were not obtained in this study. The low hemoglobin and platelet count at the time of the CMV antigenemia test reflect the hematological effects of CMV viremia and/or infection. In this study, two patients developed thrombocytopenia, one developed colitis, one developed pneumonitis, and one developed fever and anemia, indicating that CMV infection causes solid organ injuries as well as hematological disorders.

As for the method for the detection of CMV viremia and/or infection using blood samples, there are mainly three ways: evaluation of CMV-specific IgM antibody and IgG antibody, evaluation of CMV-pp65-antigen-positive cell count, and detection of CMV-DNA using PCR [26,27]. The evaluation of CMV-specific IgM antibody and IgG antibody is not effective in AAV patients. Because AAV patients are treated with strong immunosuppressive therapy, responses of immunoglobulin are weak, and we cannot effectively detect CMV viremia and/or infection. As for CMV-pp65-antigen-positive cell count, we can detect the amount of leukocytes that are infected by CMV, and CMV-pp65-antigen-positive cell count has been widely used to detect CMV viremia and infection. However, the problem with the CMV-pp65-antigen-positive cell count is that when the AAV patients presented leukocytopenia due to, for example, cyclophosphamide therapy, we could not detect CMV viremia and/or infection effectively. In addition, the detective sensitivity of CMV-pp65-antigen-positive cell count may be low. The detective sensitivity of CMV-pp65-antigen-positive cell count is significantly lower than that of PCR for CMV-DNA [28]. Therefore, ideally, PCR for CMV-DNA may be the best way to early detect CMV viremia and/or infection; however, this test was not available under insurance coverage during the current study period in Japan. As we demonstrated, the CMV antigenemia test is useful for the detection of CMV viremia leading to a CMV infection; not all CMV infections present with CMV viremia. CMV reactivation and infection often lead to lung, liver, retina, and/or colon injuries as well as hematological disorders; however, it has been reported that a CMV-pp65-antigen-positive cell count was not detected in some patients who developed CMV infection, specifically patients who developed CMV colitis [29,30]. Therefore, CMV infection should always be included in the differential diagnosis of patients with lung lesions, liver dysfunction, abdominal symptoms, eye symptoms, and/or hematological disorders during the treatment of AAV.

All patients diagnosed with CMV viremia improved throughout the study period. The dose of PSL was reduced in five patients, seven patients received ganciclovir or valganciclovir, and five patients did not require any treatment. Patients with CMV-pp65-antigen-positive cell counts < 6 were observed and/or their PSL dose was reduced. Patients with CMV-pp65-antigen-positive cell counts > 5 and those diagnosed with CMV infection were treated with ganciclovir or valganciclovir. The treatment patterns of patients in this study are similar to those of previous studies [31,32]. Morishita et al. also initiated preemptive therapy against CMV viremia when the CMV-pp65-antigen-positive cell count was >5 [6]. Saracino et al. reported that a CMV-pp65-antigen-positive cell count/white blood cell count ratio ≥ 2/2,000,000 may be an appropriate threshold at which to start anti-CMV therapy in kidney transplant recipients [31]. Gondo et al. reported that a CMV-pp65-antigen-positive cell count/white blood cell count ratio > 10/50,000 (HRP-C7 method) increased the risk of CMV infection in bone marrow transplant patients [32]. Thus, the CMV-pp65-antigen-positive cell count may be useful when determining the treatment strategy of patients with CMV viremia. A properly timed antigenemia test and evaluation of the CMV-pp65-antigen-positive cell count may lead to a good prognosis for patients with CMV viremia and/or infection.

This study has several limitations. This was a single-center, retrospective, observational study with a small sample size that prohibited the use of multivariate analyses. This study was performed in the nephrology department, and therefore, most patients presented with rapid progressive glomerulonephritis. Therefore, the characteristics of the included patients may be biased to patients with severe kidney dysfunction. In addition, as for the timing of the screening test of CMV viremia or infection, no established criteria existed for screening CMV; however, treating physicians decided to perform screening tests of CMV viremia and infection. However, our study may be useful in considering the kind of AAV patients that should be checked with CMV screening during their clinical course in the future. Furthermore, the treatment pattern against CMV viremia was not strictly controlled; however, it was based on the management of CMV viremia in kidney transplant patients. This study does not include any information regarding CMV during the maintenance phase of immunosuppressive therapy for AAV. Further research is necessary to determine the prevalence and risk factors of CMV viremia or infection in patients undergoing maintenance therapy for AAV.

## 5. Conclusions

As anti-CMV-IgG has been detected in a high percentage of the elderly population and patients with AAV, CMV reactivation leading to CMV viremia or infection can occur. The total protein, hemoglobin, platelet count and lymphocyte count are useful markers for the prediction of CMV viremia and infection after the start of induction of immunosuppressive therapy for AAV. 

## Figures and Tables

**Figure 1 jcm-12-00351-f001:**
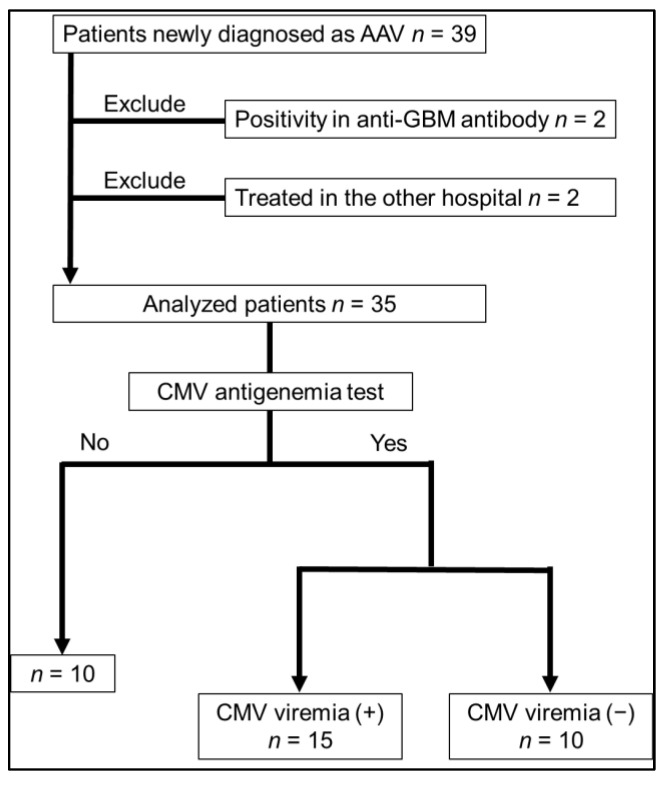
Study flow. AAV, antibody-associated vasculitis; CMV, cytomegalovirus.

**Figure 2 jcm-12-00351-f002:**
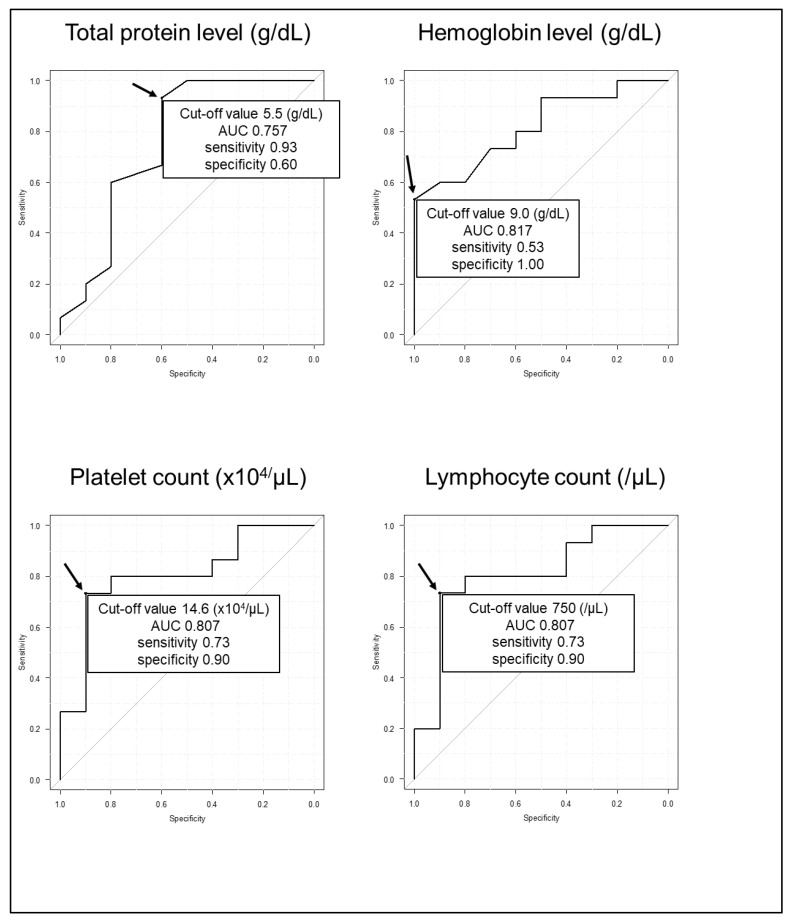
Receiver operating curve analyses and the cutoff values for predicting CMV viremia. The cutoff values for detecting CMV viremia of total protein, hemoglobin, platelet count and lymphocyte count are 5.5 g/dL (area under curve (AUC): 0.757; 95% confidence interval (CI): 0.531–0.983; Sensitivity: 0.93; Specificity: 0.60), 9.0 g/dl (AUC: 0.817; 95% CI: 0.651–0.982; Sensitivity: 0.53; Specificity: 1.00), 14.6 × 10^4^/μL (AUC: 0.807; 95% CI: 0.623–0.991; Sensitivity: 0.73; Specificity: 0.90) and 750/μL (AUC: 0.807; 95% CI: 0.618–0.995; Sensitivity: 0.73; Specificity: 0.90), respectively. The arrows point to the cutoff points.

**Figure 3 jcm-12-00351-f003:**
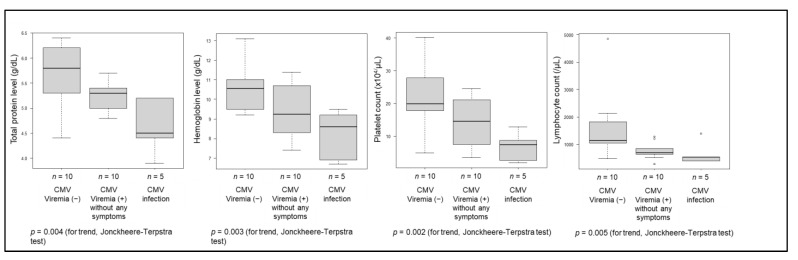
Trends of total protein, hemoglobin, platelet count, and lymphocyte count at the time of the CMV antigenemia test.

**Table 1 jcm-12-00351-t001:** The background characteristics, laboratory data, treatment pattern and clinical events of all patients and comparison between patients who received CMV antigenemia test or not.

	All Patients		Patients Who Were Received		Patients Who Were Not Received		
			CMV Antigenemia Test		CMV Antigenemia Test		*p* Value
	n = 35		n = 25		n = 10		
Age	74	47–89	72	47–89	80	63–85	0.36
Male (n, %)	20	57.1	12	48.0	8	80.0	0.13
BMI (kg/m^2^)	22.1	15.1–33.4	22.1	15.1–30.4	21.4	18.3–33.4	0.90
Systolic BP (mmHg)	139	82–205	146	82–190	135	95–205	0.62
Diastolic BP (mmHg)	76	55–115	81	55–115	74	63–91	0.78
Heart rate (/min)	75	49–115	79	56–115	65	49–82	* 0.022
Diabetes mellitus (n, %)	10	28.6	5	20.0	5	50.0	0.11
Hypertension (n, %)	24	68.6	18	72.0	6	60.0	0.69
Smoking history (n, %)	19	54.3	11	44.0	8	80.0	0.07
Malignancy (n, %)	8	22.9	7	28.0	1	10.0	0.39
Interstitial pneumonitis (n, %)	19	54.3	14	56.0	5	50.0	1.00
Alveolar hemorrhage (n, %)	6	17.1	6	24.0	0	0.0	0.15
Neurological disorder (n, %)	4	11.4	2	8.0	2	20.0	0.56
Purpura (n, %)	3	8.6	2	8.0	1	10.0	1.00
Slight fever (38 °C>, ≥37 °C) (n, %)	8	22.9	7	28.0	1	10.0	0.39
Clinical classification							
GPA (n, %)	0	0	0	0.00	0	0.0	-
MPA (n, %)	33	94.2	24	96.0	9	90.0	0.50
EGPA (n, %)	1	2.9	1	4.0	0	0.0	1.00
Unclassifiable (n, %)	1	2.9	0	0.0	1	10.0	0.32
Clinical severity (BVAS-3)	18	11–26	18	11–26	18	12–22	0.16
CMV antigenemia test (n, %)	25	71.4	25	100.0	0	0.0	-
Laboratory data							
Total protein (g/dL)	6.4	4.5–8.1	6.3	4.5–7.6	6.6	5.5–8.1	0.19
Albumin (g/dL)	2.9	1.1–4.0	2.7	1.1–3.6	3.1	1.7–4.0	0.18
Blood urea nitrogen (mg/dL)	47.4	2.2–138.3	64.8	2.2–138.3	33.9	13.0–70.6	* 0.014
Creatinine (mg/dL)	3.38	0.69–10.56	4.04	0.77–10.56	2.21	0.69–10.27	* 0.028
C-reactive protein (mg/dL)	2.89	0.02–25.26	4.25	0.03–25.26	0.58	0.02–22.50	0.12
White blood cell count (/µL)	7050	3100–48,280	7900	3100–48,280	6550	4270–17,280	0.16
Neutrophil count (/µL)	5584	1866–46,832	5772	1866–46,832	4194	3160–14,446	0.14
Lymphocyte count (/µL)	1056	0–2322	984	0–1963	1219	570–2322	0.32
Hemoglobin (g/dL)	10	6.5–13.6	9.1	6.5–12.3	10.8	8.4–13.6	* 0.037
Platelet count (×10^4^/µL)	30.3	8.3–59.4	30.3	8.3–59.2	28.6	18.6–41.4	0.86
Hematuria (n, %)	34	97.1	25	100.0	9	90.0	0.29
Proteinuria (n, %)	35	100.0	25	100.0	10	100.0	-
Positivity in MPO-ANCA (n, %)	35	100.0	25	100.0	10	100.0	-
Positivity in PR3-ANCA (n, %)	0	0.0	0	0.0	0	0.0	-
Positivity in CMV-IgG (n, %)	35	100.0	25	100.0	10	100.0	-
Treatment pattern							
PSL (maximum) (mg/kg/day)	0.72	0.52–1.00	0.74	0.63–0.85	0.62	0.52–0.83	* 0.045
Rapid PSL reduction (n, %)	15	42.9	13	52.0	2	20.0	0.13
mPSL pulse (n, %)	28	80.0	22	88.0	6	60.0	0.16
Cyclophosphamide (n, %)	7	20.0	7	28.0	0	0.00	0.08
Rituximab (n, %)	6	17.1	6	24.0	0	0.00	0.15
Plasma exchange (n, %)	7	20.0	7	28.0	0	0.00	0.08
TMP-SMX (n, %)	34	97.1	24	96.0	10	100.0	1.00
Clinical events							
CMV infection (n, %)	5	14.3	5	20.0	0	0.00	0.29
ESRD (n, %)	8	22.9	7	28.0	1	10.0	0.39
All cause of hospital death (n, %)	1	2.9	1	4.0	0	0.00	1.00

Continuous variables are presented as median and range and categorical variables as number (n) and percentages (%). The comparison of continuous variables was performed with Mann–Whitney U test, and the comparison of categorical variables was performed with Fisher’s exact probable test. A *p* value < 0.05 was considered statistically significant (presented with asterisk *). ANCA: anti-neutrophil cytoplasmic antibody, BP: blood pressure, BVAS: Birmingham vasculitis activity score, CMV: cytomegalovirus, EGPA: eosinophilic granulomatous with polyangiitis, ESRD: end stage renal disease, GPA: granulomatous with polyangiitis, IgG: immunoglobulin G, MPA: microscopic polyangiitis, MPO: myeloperoxidase, mPSL: methylprednisolone, PR3: proteinase 3, PSL: prednisolone, TMP-SMX: trimethoprim-sulfamethoxazole.

**Table 2 jcm-12-00351-t002:** Comparison of the background characteristics, laboratory data, treatment pattern and clinical events between patients who developed CMV and those who did not.

	CMV Viremia (+)		CMV Viremia (−)		*p* Value
	n = 15		n = 10		
Age	70	47–89	73	67–88	0.47
Male (n, %)	7	46.7	5	50.0	1.00
BMI (kg/m^2^)	22.5	17.6–30.4	20.1	15.1–24.4	0.08
Systolic BP (mmHg)	141	113–186	150	82–190	1.00
Diastolic BP (mmHg)	81	61–115	77	55–104	0.58
Heart rate (/min)	81	56–93	75	60–115	0.93
Diabetes mellitus (n, %)	3	20.0	2	20.0	1.00
Hypertension (n, %)	11	73.3	7	70.0	1.00
Smoking history (n, %)	6	40.0	5	50.0	0.70
Malignancy (n, %)	6	40.0	1	10.0	0.18
Duration of follow-up (day)	25	5–44	27.5	13–95	0.56
Interstitial pneumonitis (n, %)	8	53.3	6	60.0	1.00
Alveolar hemorrhage (n, %)	5	33.3	1	10.0	0.35
Neurological disorder (n, %)	2	13.3	0	0.0	0.50
Purpura (n, %)	2	13.3	0	0.0	0.50
Clinical severity (BVAS-3)	20	12–26	18	11–21	0.21
Slight fever (38 °C>, ≥37 °C) (n, %)	6	40.0	1	10.0	0.18
Laboratory data (at the admission)					
Total protein (g/dL)	6.3	4.5–7.4	6.6	4.6–7.6	0.23
Albumin (g/dL)	2.6	1.1–3.6	3.0	1.6–3.6	0.17
IgG (mg/dL)	1500	758–2269	1390	787–1906	0.40
Blood urea nitrogen (mg/dL)	75.4	13.9–138.3	45.1	2.22–77.9	* 0.031
Creatinine (mg/dL)	4.47	0.92–10.56	3.38	0.77–8.63	0.26
C-reactive protein (mg/dL)	6.10	0.03–25.26	1.84	0.05–17.32	0.45
White blood cell count (/µL)	10,670	6070–48,280	5915	3100–22,820	* 0.002
Neutrophil count (/µL)	8422	5020–46,832	4264	1866–20,196	* 0.004
Lymphocyte count (/µL)	710	0–1683	1070	601–1963	0.18
Hemoglobin (g/dL)	9.3	6.5–12.3	8.3	7.2–11.2	0.52
Platelet count (×10^4^/µL)	24.0	8.3–59.4	32.8	15.8–53.3	0.20
Hematuria (n, %)	15	100.0	10	100.0	-
Proteinuria (n, %)	15	100.0	10	100.0	-
Positivity in MPO-ANCA (n, %)	15	100.0	10	100.0	-
Positivity in PR3-ANCA (n, %)	0	0.0	0	0.0	-
Positivity in CMV-IgG (n, %)	15	100.0	10	100.0	-
High titer of CMV-IgG (n, %)	4	26.7	6	60.0	0.12
Treatment pattern					
PSL (maximum) (mg/kg/day)	0.78	0.56–1.00	0.66	0.54–0.88	0.10
PSL (maximum) (mg/day)	45	25–70	30	20–50	0.07
Rapid PSL reduction (n, %)	9	60.0	4	40.0	0.43
mPSL pulse (n, %)	15	100.0	7	70.0	0.05
Cyclophosphamide (n, %)	4	26.7	3	30.0	1.00
Rituximab (n, %)	5	33.3	1	10.0	0.34
Plasma exchange (n, %)	6	40.0	1	10.0	0.18
PSL (alone)	0	0.0	3	30.0	0.05
PSL + CY	1	6.7	0	0.0	1.00
PSL + mPSL	8	53.3	3	30.0	0.41
PSL + mPSL + CY	2	13.3	3	30.0	0.36
PSL + mPSL + RIT	3	20.0	1	10.0	0.63
PSL + mPSL + CY + RIT	1	6.7	0	0.0	1.00
ST mixture (n, %)	15	100.0	9	90.0	0.40
Clinical events					
CMV infection (n, %)	5	33.3	0	0.0	0.06
ESRD (n, %)	4	26.7	3	30.0	1.00
All cause of hospital death (n, %)	0	0.0	1	10.0	0.40

Continuous variables are presented as median and range and categorical variables as number (n) and percentages (%). The comparison of continuous variables was performed with Mann–Whitney U test, and the comparison of categorical variables was performed with Fisher’s exact probable test. A *p* value < 0.05 was considered statistically significant (presented with asterisk *). ANCA: anti-neutrophil cytoplasmic antibody, BP: blood pressure, BVAS: Birmingham vasculitis activity score, CMV: cytomegalovirus, CY: cyclophosphamide, EGPA: eosinophilic granulomatous with polyangiitis, ESRD: end stage renal disease, GPA: granulomatous with polyangiitis, IgG: immunoglobulin G, MPA: microscopic polyangiitis, MPO: myeloperoxidase, mPSL: methylprednisolone, PR3: proteinase 3, PSL: prednisolone, RIT: rituximab, TMP-SMX: trimethoprim-sulfamethoxazole.

**Table 3 jcm-12-00351-t003:** Analyses of the association between CMV viremia and background clinical factor and treatment pattern.

	OR	95% CI	*p* Value
Age	0.97	0.90–1.04	0.36
Male (n, %)	0.88	0.18–4.34	0.87
BMI (kg/m^2^)	1.38	0.96–1.99	0.08
Diabetes mellitus (n, %)	1.00	0.14–7.39	1.00
Clinical severity (BVAS-3)	1.21	0.94–1.55	0.15
White blood cell count (/µL)	1.00	1.00–1.00	0.10
Neutrophil count (/µL)	1.00	1.00–1.00	0.34
Lymphocyte count (/µL)	0.99	0.99–1.00	0.16
CRP (mg/dL)	1.07	0.92–1.23	0.40
High titer of CMV-IgG (n, %)	0.24	0.04–1.33	0.10
High dose PSL (maximum) (mg/kg/day)	3.50	0.64–19.2	0.15
CY use	0.85	0.14–4.99	0.86
RIT use	4.50	0.44–46.2	0.21

Univariate logistic regression analyses were performed. High titer of anti-CMV-IgG was defined as an anti-CMV-IgG titer of more than 250 AU/mL. High dose of PSL was defined as the patients who received PSL more than median dose. BVAS: Birmingham vasculitis activity score, CI: confidence interval, CMV: cytomegalovirus, CY: cyclophosphamide, IgG: immunoglobulin G, OR: odds ratio, PSL: prednisolone, RIT: rituximab.

**Table 4 jcm-12-00351-t004:** The comparison of the laboratory data between patients who developed CMV viremia and those who did not (at the time of the CMV antigenemia test).

	CMV Viremia (+)		CMV Viremia (−)		*p* Value
	n = 15		n = 10		
Duration between the start of immunosuppressive therapy and CMV antigenemia test (day)	23.5	9–36	25	9–95	0.98
Laboratory data					
(at the time of CMV antigenemia test)					
Total protein (g/dL)	5.2	3.9–5.7	5.8	4.4–6.4	* 0.035
Albumin (g/dL)	2.8	2.2–3.3	2.9	1.8–4.2	1.00
Blood urea nitrogen (mg/dL)	50.5	19.2–99.6	49.0	16.5–148.2	1.00
Creatinine (mg/dL)	3.45	1.37–7.73	2.83	0.87–6.64	0.90
Aspartate aminotransferase (U/L)	15	8–39	17	5–110	0.66
Alanine transaminase (U/L)	15	6–50	22	3–208	0.93
Alkaline phosphatase (U/L)	204	120–389	216	139–1309	0.23
γ-Glutamyl transpeptidase (U/L)	21	14–68	33	13–1020	0.15
Total bilirubin (mg/dL)	0.57	0.25–0.91	0.43	0.29–0.91	0.27
C-reactive protein (mg/dL)	0.16	0.02–1.85	0.12	0–4.08	0.85
White blood cell count (/µL)	5680	2690–15,280	7565	5250–19,590	0.08
Neutrophil count (/µL)	4586	1778–13,614	5622	4200–13,537	0.13
Lymphocyte count (/µL)	673	283–1390	1142	488–4858	* 0.01
Hemoglobin (g/dL)	9	6.7–11.4	10.6	9.2–13.1	* 0.009
Platelet count (×10^4^/µL)	12.9	2.0–24.5	20.0	5.0–40.2	* 0.012
Hematuria (n, %)	15	100.0	8	80.0	0.15
Proteinuria (n, %)	15	100.0	10	100.0	-

Continuous variables are presented as median and range and categorical variables as number (n) and percentages (%). The comparison of continuous variables was performed with Mann–Whitney U test, and the comparison of categorical variables was performed by Fisher’s exact probable test. A *p* value < 0.05 was considered statistically significant (we presented with asterisk *). CMV: cytomegalovirus.

**Table 5 jcm-12-00351-t005:** Association between laboratory data at the time of CMV antigenemia test and CMV viremia.

	OR	95% CI	*p* Value
Total protein (g/dL)	0.13	0.02–0.90	* 0.038
Albumin (g/dL)	0.84	0.16–4.27	0.83
Blood urea nitrogen (mg/dL)	1.00	0.97–1.02	0.84
Creatinine (mg/dL)	0.99	0.65–1.51	0.96
Aspartate aminotransferase (U/L)	0.98	0.93–1.03	0.35
Alanine transaminase (U/L)	0.99	0.96–1.02	0.36
White blood cell count (/µL)	1.00	1.00–1.00	0.32
Lymphocyte count (/µL)	0.99	0.99–1.00	* 0.028
C-reactive protein (mg/dL)	0.53	0.22–1.26	0.15
Hemoglobin (g/dL)	0.37	0.15–0.89	* 0.027
Platelet count (×10^4^/µL)	0.86	0.75–0.98	* 0.027

Univariate logistic regression analyses performed.* *p* value < 0.05 considered statistically significant. CI: confidence interval, OR: odds ratio.

**Table 6 jcm-12-00351-t006:** Clinical course of AAV patients who developed CMV viremia after the start of induction of immunosuppressive therapy.

Age	Sex	BVAS	Maximum Dose of PSL (mg/kg/day)	Details of the Induction Immunosuppressive Therapy	Days between the Start of Induction Therapy to the Detection of CMV Viremia	Maximum Value CMV pp65 Antigen Positive Cell Count	CMV Infection	Details of the Intervention against CMV Viremia	Prognosis of CMV Viremia
89	F	12	0.59	PSL 25 mg, mPSL 500 mg, RIT 500 mg	25	3	-	PSL dose reduction	Improve
55	M	16	0.74	PSL 60 mg, mPSL 1000 mg, CY 1000 mg, RIT 700 mg	30	1	-	PSL dose reduction	Improve
85	M	20	0.67	PSL 30 mg, mPSL 500 mg, RIT 500 mg	31	1	-	Observation	Improve
50	M	18	1.00	PSL 70 mg, CY 1000 mg	29	2	-	Observation	Improve
74	F	18	0.78	PSL 40 mg, mPSL 1000 mg	25	6	-	PSL dose reduction and ganciclovir	Improve
70	F	24	0.86	PSL 35 mg, mPSL 1000 mg	25	6	colitis	PSL dose reduction and valganciclovir	Improve
69	F	15	0.75	PSL 40 mg, mPSL 1000 mg, CY 500 mg	26	2	-	Observation	Improve
63	F	23	0.85	PSL 45 mg, mPSL 1000 mg, CY 700 mg	36	3	-	Observation	Improve
47	F	20	0.79	PSL 50 mg, mPSL 1000 mg	28	10	fever, anemia	Ganciclovir	Improve
87	F	18	0.67	PSL 30 mg, mPSL 500 mg	19	2	-	Observation	Improve
83	M	24	0.74	PSL 60 mg, mPSL 500 mg	9	31	thrombocytopenia	Ganciclovir	Improve
86	M	21	0.90	PSL 50 mg, mPSL 500 mg	21	10	-	Ganciclovir	Improve
78	M	20	0.85	PSL 50 mg, mPSL 500 mg	14	18	thrombocytopenia	Valganciclovir	Improve
64	M	17	0.86	PSL 50 mg, mPSL 1000 mg, RIT 200 mg	22	1	-	PSL dose reduction	Improve
61	F	26	0.56	PSL 30 mg, mPSL 1000 mg	24	296	pneumonitis	Ganciclovir, Valganciclovir	Improve

AAV: anti-neutrophil cytoplasmic antibody-associated vasculitis, BVAS: Birmingham vasculitis activity score, CMV: cytomegalovirus, CY: cyclophosphamide, F: female, M: male, mPSL: methylprednisolone, PSL: prednisolone, RIT: rituximab.

## Data Availability

The data used in this study are available from the corresponding author upon request.

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
