# Peer review of "Predictive Factors of Cytomegalovirus Viremia during the Clinical Course of Anti-Neutrophil Cytoplasmic Antibody (ANCA)-Associated Vasculitis: A Single Center Observational Study"

_jcm, 2023, doi:10.3390/jcm12010351_

Round 1

Reviewer 1 Report

The manuscript is in good shape. However, minor changes are needed.

The author should provide the latest references regarding how they are using these as antibodies.

The author needs to provide the disadvantages/side effects of this.

Author Response

Reviewer 1

The manuscript is in good shape. However, minor changes are needed.

Thank you for finding the time in your busy schedule to review our manuscript. According to your comments, we revised our manuscript.

The author should provide the latest references regarding how they are using these as antibodies.

The author needs to provide the disadvantages/side effects of this.

Thank you for your advice.

As for the method for the detection of CMV viremia and/or infection, there are mainly three ways; evaluation of CMV-specific IgM antibody and IgG antibody, evaluation of CMV-pp65-antigen-positive cell count, detection of CMV-DNA using PCR (PMID: 29596116, PMID: 32134488). The evaluation of CMV-specific IgM antibody and IgG antibody is not effective in AAV patients. Because AAV patients are treated with strong immunosuppressive therapy, responses of immunoglobulin are weak and we cannot effectively detect CMV viremia and/or infection. As for CMV-pp65-antigen-positive cell count, we can detect the leukocyte count that are infected by CMV. The problem of CMV-pp65-antigen-positive cell count is that when the AAV patients presented leukocytopenia due to such as cyclophosphamide therapy, we cannot detect CMV viremia and/or infection effectively. In addition, the detective sensitivity of CMV-pp65-antigen-positive cell count is lower than that is PCR for CMV-DNA (PMID: 28506786). Therefore, ideally, PCR for CMV-DNA may be best way to early detection of CMV viremia and/or infection, however, this test was not available in the insurance coverage during current study period in Japan.

As for the other way to predict CMV viremia and/or infection, as we discussed, Kaneshita et al. reported that oral candida infection, high CMV-pp65-antigen-positive cell counts, and hypoalbuminemia are possible risk factors for the progression of CMV viremia to CMV infection. In addition, recently, Shimada et al reported that elderly, low level of serum albumin, higher creatinine level, cyclosporine use, and higher maximum and cumulative doses of PSL were the risk factors of CMV re-activation in rheumatic diseases (PMID: 36463264). Suga et al. reported that age > 60 years, lymphocytopenia (< 1000/μL) and steroid pulse therapy were the risk factors of CMV infection in rhematic disease (PMID: 36042051).

As for the lymphocytopenia, we also detected as a predictive factor of CMV viremia and its cut-off value was <750/μL in the current study.

We added these comments to discussion section (Page 15, lines 345-362, and page 14, lines 303-309).

Reviewer 2 Report

the idea of viral infections in patients with autoimmune diseases is interesting and insufficiently studied. 

The authors should describe in the methodology the reasons that led to the screening of CMV. The clinical symptoms and the laboratory tests (eg lymphocyte count) should be described. 

A matched group of patients can be used for comparison. 

The sample size is rather limited to extract conclusions. 

Author Response

Reviewer 2

the idea of viral infections in patients with autoimmune diseases is interesting and insufficiently studied.

Thank you for finding the time in your busy schedule to review our manuscript. According to your comments, we have tried to respond to all of your concerns and improve our manuscript as much as possible. However, because the due date was December 10th, it is possible that we were unable to fully address all of your queries in time.

The authors should describe in the methodology the reasons that led to the screening of CMV.

As for the timing of the screening test of CMV viremia or infection, because this was a retrospective study and no established criteria that when we need to screen CMV in AAV patients was existed, treating physicians decided to perform screening test of CMV viremia and infection. Therefore, this is the limitation of the current study. We added these comments to discussion as a limitation. However, our study may be useful to consider what kind of AAV patients should be checked CMV screening in their clinical course. (Page 16, lines 391- 396)

The clinical symptoms and the laboratory tests (eg lymphocyte count) should be described.

As for the clinical symptoms, of 35 patients, no patients were detected to present obvious fever (>38℃), and 8 patients had slight fever (38℃>, ≥37℃) at the time of hospital admission. The other symptoms such as purpura and neurological disorder are presented in the tables. As for the clinical symptoms among patients who developed CMV viremia, they are asymptomatic except for patients who developed CMV infection (page 5, lines 162-166, page 6, lines 184-186).

As for the laboratory test, we added “lymphocyte count and neutrophil count” at the hospital admission and at the time of evaluating CMV antigenemia test to table 1 and 2. In addition, we also performed the same regression analyses as other clinical factors to investigate the association with CMV viremia. As a result, we found that lymphocyte count at the timing of CMV antigenemia test was also significantly associated with the CMV viremia. We added these results to appropriate tables, figures, figure legends and result section, respectively (page 6, lines 189-192, page 8, lines 207-210, 212-213, page 10, lines 234-243, page 13, lines 274-276, and page 16, lines 391-396).

A matched group of patients can be used for comparison.

Unfortunately, we do not have so much patients with AAV because this is single center observational research and AAV is rare disease. The validation study of investigating the efficacy of our current result using another cohort will be needed, and we are investigating whether the total protein, hemoglobin, platelet count and lymphocyte count are useful marker for the prediction of CMV viremia or not in the further cohort. Now we are accumulating the cases of AAV patients.

The sample size is rather limited to extract conclusions.

As we mentioned above query, because this is single center observational research and as you know AAV is originally rare disease, it is difficult to collect many numbers of patients with AAV. Although you suggested that the sample size was low and it may have led to this study being statistically underpowered, we presented the detailed clinical characteristics and course of patients who developed CMV viremia in the table, instead. We believe that this information is useful for the clinical physicians who treat AAV patients.